# Long-Term Change of a Fish-Based Index of Biotic Integrity for a Semi-Enclosed Bay in the Beibu Gulf

Li Su [1,2], Youwei Xu [1,2,3], Yongsong Qiu [1,2], Mingshuai Sun [1,2], Kui Zhang [1,2] and Zuozhi Chen [1,2,3,4,*]

1   South China Sea Fisheries Research Institute, Chinese Academy of Fishery Sciences, Guangzhou 510300, China; suli@scsfri.ac.cn (L.S.); xuyouwei@scsfri.ac.cn (Y.X.); qys@scsfri.ac.cn (Y.Q.); sunmingshuai@scsfri.ac.cn (M.S.); zhangkui@scsfri.ac.cn (K.Z.)
2   Key Laboratory for Sustainable Utilization of Open-Sea Fishery, Ministry of Agriculture and Rural Affairs, Guangzhou 510300, China
3   Southern Marine Science and Engineering Guangdong Laboratory (Guangzhou), Guangzhou 511458, China
4   Guangdong Provincial Key Laboratory of Fishery Ecology and Environment, Guangzhou 510300, China
*   Correspondence: chenzuozhi@scsfri.ac.cn

**Abstract:** The Beibu Gulf in the northwestern South China Sea is one of the world's most productive fishing grounds, yet its biotic resources appear to be in rapid decline because of overexploitation. Assessments of the health status of the fisheries' resources in the gulf provide a foundation for their conservation and management. As fish accounted for 84% to 97% of the total catch in the Beibu Gulf in the period 1962–2017, a Fish-based Index of Biotic Integrity (F-IBI) was developed for the Beibu Gulf, and data from otter trawl surveys during the period 1962–2017 were used to measure variations in the fish community. The assessment revealed a generally downward trend in total fish catch density ($kg/km^2$), catch density of 12 traditional commercial demersal fish species, and percentage of nektonic-feeding species, but an upward trend for the percentage of fishes with a planktivorous or detritus diet, percentage of pelagic species, and percentage of the dominant species. The dominant species varied greatly over the 50 year period and showed a tendency towards small-sized species. The synthetic F-IBI variable showed a downward trend and has indicated a 'fair' state since 1998. The decline in the F-IBI over the last three decades suggests that anthropogenic disturbances, especially overfishing, have had a serious impact on the fish community of the Beibu Gulf. The F-IBI is currently at risk of becoming 'poor.' Consequently, we suggest that rigid and enforceable fishery management measures should be taken by both China and Vietnam to prevent further deterioration of the fisheries' resources in the gulf.

**Keywords:** fish community; index of biotic integrity; fisheries resource; management; Beibu Gulf





## 1. Introduction

Fish are a vital component of aquatic ecosystems and have adapted to a wide range of habitats in the long course of historical evolution, resulting in extremely varied morphological, ecological, and physiological characteristics. The marine environment has the largest number of fish species. From 1980 to 2016, fish were the main component of marine catch in Guangdong Province, accounting for 75.85% of the average annual catch [1]. The Beibu Gulf is a semi-enclosed bay located in the northwest of the South China Sea (17–21.75° N, 105.67–110.17° E), bordering China's Leizhou Peninsula and Hainan Island to the east, Guangxi Zhuang Autonomous Region to the north, and Vietnam to the west (Figure 1). With unique geographical and climatic conditions, it is rich in fishery resources and is one of China's four most prominent fishing grounds. In recent years, the number and power of fishing boats entering the Beibu Gulf from China and Vietnam have maintained or increased [2]. Due to long-term intensive fishing, the fishery resources in this area have seriously declined and the community has also changed greatly. In particular, demersal catch composition showed a changing trend from higher quality, high trophic level, and

larger-size fishes to lower value, low trophic level, and smaller-size ones [3,4]. How to better evaluate the decline in resources is an urgent problem to be solved in order to effectively guide the development and conservation of the fisheries.

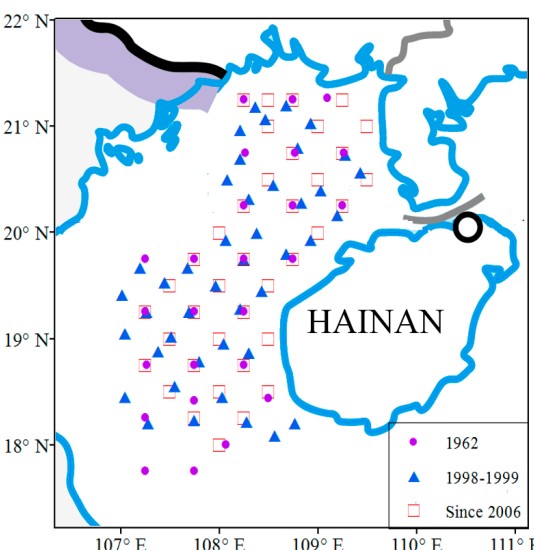

**Figure 1.** Map of Beibu Gulf and sampling stations of the fishery stock survey used in the present analysis (dots).

Karr [5] first introduced the concept an Index of Biotic Integrity (IBI) to assess the resource quality of small warmwater streams in the Midwestern United States based on the attributes of the fish community. Since then, IBIs have been developed and applied in many other regions and countries, such as elsewhere in the United States [6,7], France [8], New Zealand [9], Lithuania [10], Africa [11], Argentina [12], Brazil [13,14], and Mexico [15]. In China, many studies have applied an IBI to assess the conditions of rivers, estuaries, and lakes [16–19]. Lin et al. [20] developed an IBI for a bay to assess the fish community status, and the results showed that the level of fish biological integrity in this area was poor. The IBI is a useful biomonitoring tool because it infers the quality of environments by considering different levels of fish community organization [7,13]. Furthermore, use of the index can help to elucidate specific aspects of ecosystem degradation, to formulate and apply management strategies for mitigation or restoration, and to evaluate the success of these strategies [21].

The purpose of this study was to develop an IBI index suitable for assessing the changes in the integrity of the fishery resources in the Beibu Gulf in different years. The index scores were developed based on trawl surveys conducted for 5 years over the period 1962 to 2017. The temporal trends in the IBI scores were determined and the possible causes of change were analyzed. This study provides a reference for ecosystem health assessment, sustainable utilization, conservation, and fishery resource management in the Beibu Gulf.

## 2. Materials and Methods

### 2.1. Fish Samples

Surveys of fishery resources in the Beibu Gulf were conducted by the South China Sea Fisheries Research Institute in 1962, 1998–1999, 2006, 2014–2015, and 2016–2017. All surveys were carried out according to the 'Specifications of Oceanographic Survey'; using otter trawls for sampling, catches at each sampling station were sorted to species level, enumerated, and weighed. To use the catch data from the different sampling periods, we determined an area of approximate overlap covered by all the surveys and used quarterly sampling data from 25 to 43 sites for the present analysis (Figure 1; Table 1). Fish abundance was expressed as catch rate (catch per unit effort, CPUE). However, the vessels, trawl gears, and towing speeds used in the 1962 and 1998–1999 surveys varied from those of the surveys

since 2006. To make the catch rates derived from the 1962 and 1998–1999 data comparable to the data collected since 2006, we calibrated CPUE as catch density ($D$, kg/km$^2$) according to the formula:

$$D = C/(v \cdot t \cdot L \cdot X)$$

where, $C$ is the catch biomass (kg); $v$ is the otter trawl speed (km/h); $t$ is the trawling duration (h); $L$ is the headrope length (km) of an otter trawl net; and $X$ is the fraction of headrope length of a trawl net, with 0.66 used for this study [22]. We also referred to FishBase (https://www.fishbase.de) to obtain information on food preferences and feeding habits of each fish species.

**Table 1.** Details of the fishery stock surveys in the Beibu Gulf.

| Year | 1962 | 1998–1999 | 2006 | 2014–2015 | 2016–2017 |
|---|---|---|---|---|---|
| Vessel used | Xianfeng | R/V Beidou | Beiyu60011 | Beiyu60011 | Beiyu60011 |
| Headrope length (m) | 31 | 42.8 | 37.7 | 37.7 | 37.7 |
| Cod end mesh (mm) | 40 | 20 | 39 | 39 | 39 |
| Towing speed (knot) | 2.5–3.0 | 2.5–3.5 | 3.0–4.0 | 3.0–4.0 | 3.0–4.0 |
| Sampling duration (min) | 60 or 120 | 60 | 60 | 60 | 60 |
| Station × frequency | 25 × 4 | 43 × 4 | 38 × 4 | 38 × 4 | 38 × 4 |
| Sample date | Feb., May., Aug., and Nov., 1962 | Jan., Aug., and Nov., 1998; Apr., 1999 | Jan., Apr., Jul., and Oct., 2006 | Jul. and Oct., 2014; Jan. and Apr., 2015 | Jul. and Oct., 2016; Jan. and Apr., 2017 |

### 2.2. F-IBI Estimation

A multimetric fish-based IBI (F-IBI) was used to assess the biotic integrity of the fishery resources in the Beibu Gulf. Eight metrics representing four classes of attributes—species richness, trophic composition, fish abundance and community composition, and the traditionally fished commercial demersal species were determined (Table 2). For estimations of the last two metrics, we used the data on 12 commercial demersal species: *Lutjanus sanguineus*, *Nemipterus bathybius*, *N. virgatus*, *Priacanthus macracanthus*, *P. tayenus*, *Saurida undosquamis*, *S. tumbil*, *Trichiurus lepturus*, *Argyrosomus argentatus*, *Evynnis cardinalis*, *Upeneus moluccensis*, and *U. japonicus*. These species have relatively high abundance and economic value in the Beibu Gulf.

**Table 2.** Metrics for fish integrity biotic index (F-IBI) and their expected responses to interference.

| Category | Metrics | Expected Response to Interference |
|---|---|---|
| Species richness | M1. Total number of species | decrease |
| Trophic level | M2. Percentage of density of nektonic diet species | decrease |
| | M3. Percentage of density of planktivorous and detritus diet species | increase |
| Fish abundance and compositions | M4. Total annual averaged density of fishes | decrease |
| | M5. Percentage of density of pelagic species | increase |
| | M6. Percentage of density of dominant species | increase |
| Traditional demersal commercial species | M7. Summed density of 12 traditional demersal commercial species | decrease |
| | M8. Percentage of 12 traditional demersal commercial species | decrease |

Owing to the light exploitation of the fishery stocks in the 1960s, the ecosystem during that period was relatively stable [23,24]. Therefore, 1962 conditions were considered the reference point and each metric was scored again for the 1962 data. For metrics that were expected to decrease in response to interference (i.e., fishing pressure), $S = V_{\text{Mi}}/V_{\text{M1962}}$, and for metrics expected to increase, $S = V_{\text{M1962}}/V_{\text{Mi}}$, where $V_{\text{M1962}}$ is the metric value in 1962, and $V_{\text{Mi}}$ is the value for the $i$ year. When the ratio was >1, it was assumed equal to 1, and then the scores of the eight metrics were summed to produce an IBI score ranging from a minimum of 0 to a maximum of 8. The evaluation results were then rated as excellent (6.4–8), good (4.8–6.4), fair (3.2–4.8), poor (1.6–3.2), or very poor (0–1.6). The values of F-IBI

and the metrics were plotted against the sequence of years, and the statistical significance of their trend over the year was tested using regression analysis in Excel 2016, looking at the product–moment correlation coefficient *r*.

## 3. Results

### 3.1. Dominant Species

In the 1962 survey, 450 species were recorded, and 18 species accounted for >1% of the total density of the fish caught, with demersal commercial species *L. sanguineus* having the highest proportion, followed by demersal species *Trachiocephalus myops*, *N. virgatus*, and *U. moluccensis* (Table 3). In 1998–1999, 318 species were recorded, and 18 species accounted for >1% of the total fish catch density, and the top three species were pelagic species *Siganus oramin,* the small-sized demersal species *Leiognathus bindus,* and demersal species *E. cardinalis*. In 2006, 308 species were recorded, 15 species accounted for >1% of the total fish catch density, and the top three species were small-sized demersal species *Acropoma japonicum*, pelagic species *Trachurus japonicus,* and demersal species *E. cardinalis*. In 2014–2015, 285 species were recorded, 15 species accounted for >1%, and the top three species were small-sized demersal species *A. japonicum*, demersal species *E. cardinalis,* and pelagic species *T. japonicas*. Finally, in 2016–2017, 280 species were recorded, 16 species accounted for >1%, and the top three species were small-sized demersal species *A. japonicum*, pelagic species *T. japonicus,* and pelagic species *Psenopsis anomala*. The numbers of species recorded generally decreased, and the dominant species tended to become pelagic and small-sized species.

### 3.2. F-IBI Scores

We assumed that the F-IBI was the best during 1962, with the highest value of eight. The F-IBI score highly decreased between 1962 and 1999, then slightly increased by 2006, but thereafter has shown a decreasing trend (Figure 2). The overall trend of the F-IBI was downward, with the lowest score of 3.35 in 2016–2017. According to the criteria of evaluation, the biotic integrity of the fishery resources in the Beibu Gulf could be rated as in a 'fair' state during 1998 to 2017. An analysis of trend significance found that the F-IBI against years showed a highly significant declining trend ($r = -0.920$, $p < 0.01$), despite small number of samples ($n = 5$ years).

### 3.3. Trends in the Metrics

The percentages of fish catch densities showed variable trends between two different dietary groups of fishes (Figure 3a). The percentages of nektonic-feeding species significantly decreased over the years ($r = -0.990$, $p < 0.01$), from 48.39% to 15.82%, while the percentages of species with a planktivorous or detritus diet increased from a low of 5.62% in 1962 to the highest level of 37.81% in 1998–1999, and fluctuated thereafter. It still showed a significant upward trend ($r = 0.798$, $p < 0.05$).

In terms of total fish abundance, there was a significant decline in the annual average catch density of the species ($r = -0.767$, $p < 0.05$) (Figure 3b). The highest level was 1075.93 kg/km$^2$ in 1962, decreasing to a low of 303.89 kg/km$^2$ in 1998–1999, and increasing to 643.92 kg/km$^2$ in 2006, but thereafter decreasing to 353.18 kg/km$^2$ in 2016–2017. The percentage of pelagic species in the fish catch density has significantly risen over the years ($r = 0.957$, $p < 0.01$) (Figure 3c), being lowest in 1962 and highest in 2016–2017. The percentage of the dominant species in the fish catch density also showed a significant rising trend ($r = 0.853$, $p < 0.05$), from a low of 59.61% in 1962 to 77.01% in 2016–2017.

The summed density of the 12 traditional commercial demersal fish species and their percentage in the total fish catch density, although fluctuating, also showed a significant downward trend ($r = -0.856$, $p < 0.05$; $r = -0.878$, $p < 0.01$) (Figure 3d). The highest density was 385.01 kg/km$^2$ in 1962, and the lowest was 50.51 kg/km$^2$ in 2016–2017. Their percentage in the total fish catch density declined from 35.78% in 1962 to 14.30% in 2016–2017.

**Table 3.** The species accounted for >1% of total fish catch density in the surveys.

| Species | The Proportion of Total Catch Density of Fish % | | | | |
|---|---|---|---|---|---|
| | 1962 | 1998–1999 | 2006 | 2014–2015 | 2016–2017 |
| *Lutjanus sanguineus* | 14.45 | | | | |
| *Trachiocephalus myops* | 5.90 | | | | |
| *Nemipterus virgatus* | 5.05 | 1.40 | | | |
| *Upeneus moluccensis* | 4.93 | | | | |
| *Therapon theraps* | 4.87 | | 1.90 | 1.50 | 1.31 |
| *Gerres acinaces* | 3.56 | | | | |
| *Argyrosomus macrocephalus* | 3.50 | | 3.08 | | |
| *Pomadasy hasta* | 2.39 | | | | |
| *Priacanthus macracanthus* | 2.13 | | | | |
| *Gymnocranius griseus* | 1.94 | | | | |
| *Upeneus sulphureus* | 1.86 | 2.36 | | | |
| *Caranx malabaricus* | 1.86 | | | | |
| *Nemipterus bathybius* | 1.44 | | | | |
| *Carcharhinus menisorrah* | 1.35 | | | | |
| *Abalistes stellatus* | 1.21 | | | | |
| *Decapterus maruadsi* | 1.07 | | 3.98 | 5.37 | 4.56 |
| *Arius thalassinus* | 1.05 | | | | |
| *Evynnis cardinalis* | 1.05 | 6.23 | 5.08 | 14.24 | 7.74 |
| *Siganus oramin* | | 10.81 | | | |
| *Leiognathus bindus* | | 8.43 | 4.97 | 1.46 | 1.96 |
| *Stolephorus heterlolba* | | 6.21 | | | |
| *Trichiurus brevis* | | 6.10 | | | |
| *Acropoma japonicum* | | 5.89 | 25.16 | 16.90 | 17.00 |
| *Stolephorus commersonii* | | 4.70 | | | |
| *Lutjanus johni* | | 4.44 | | | |
| *Saurida tumbil* | | 4.29 | 1.27 | 2.60 | 2.57 |
| *Stolephorus zollingeri* | | 4.28 | | | |
| *Saurida undosquamis* | | 2.58 | | | |
| *Leiognathus berbis* | | 2.32 | | | |
| *Caranx kalla* | | 1.45 | 4.12 | 1.71 | 2.91 |
| *Trichiurus lepturus* | | 1.40 | 4.31 | 2.00 | 1.04 |
| *Leiognathus lineolatus* | | 1.25 | | | |
| *Gastrophysus spadiceus* | | 1.13 | 1.41 | | |
| *Trachurus japonicus* | | | 13.23 | 12.65 | 16.76 |
| *Pennahia macrocephalus* | | | 3.74 | 3.29 | 1.03 |
| *Leiognathus ruconius* | | | 3.04 | | |
| *Apogon carinatus* | | | 1.47 | | |
| *Johnius belangerii* | | | 1.12 | 1.29 | 1.42 |
| *Apogonichthys ellioti* | | | | 4.75 | 3.67 |
| *Psenopsis anomala* | | | | 4.23 | 10.29 |
| *Gnathophis nystromi* | | | | 1.62 | 2.45 |
| *muraenesox cinereus* | | | | 1.45 | 1.31 |
| *Hippoglossoides dubius* | | | | | 1.00 |

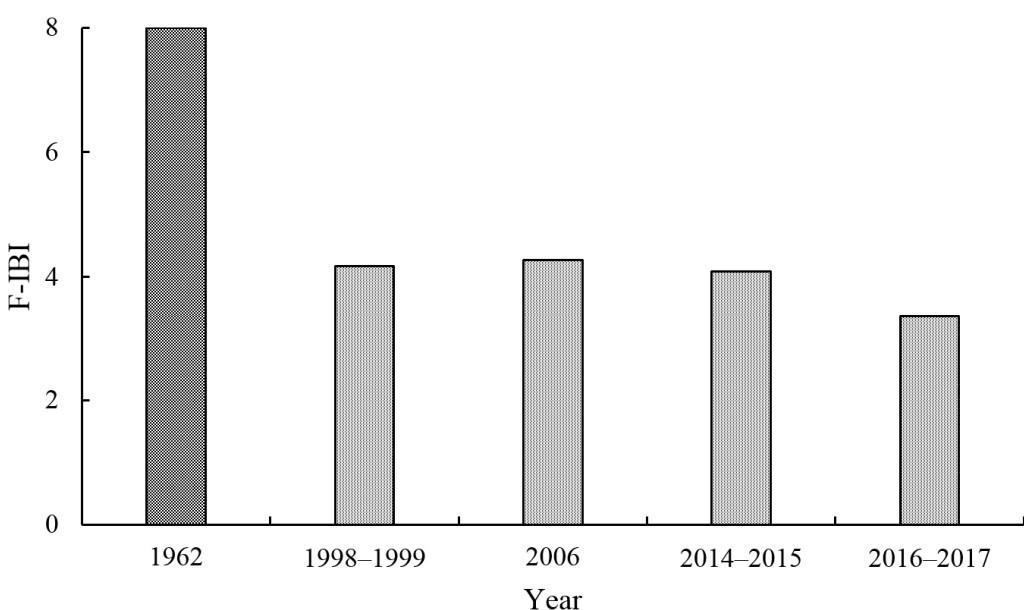

**Figure 2.** Interdecadal trend of F-IBI in the Beibu Gulf.

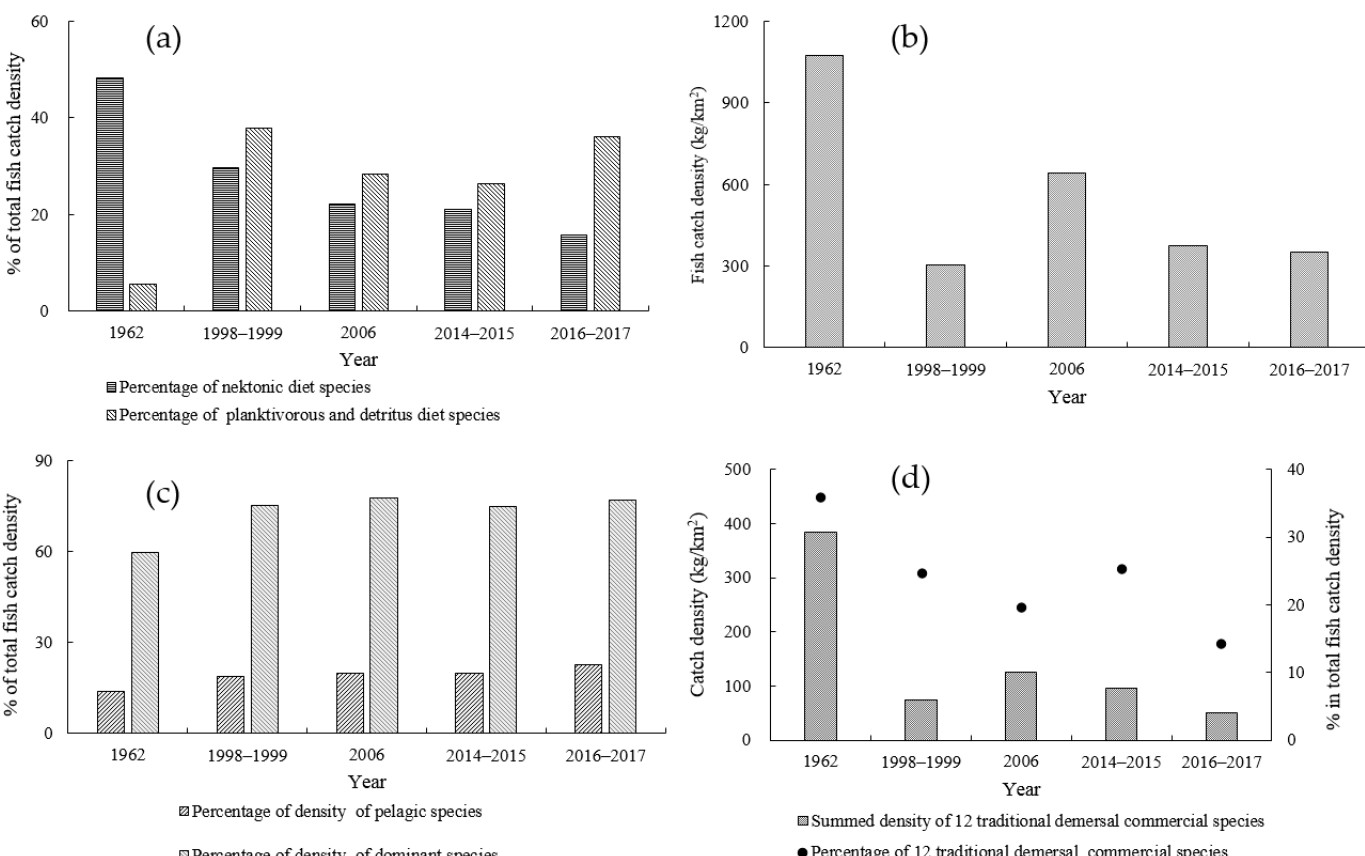

**Figure 3.** Interdecadal changes in the percentages of density of the species with nektonic diet and planktivorous and detritus diet (**a**), annual averaged catch density of fishes (**b**), the percentages of density of pelagic species and dominant species (**c**), and the summed density for the 12 traditional demersal commercial fishes (**d**) in the Beibu Gulf.

## 4. Discussion

In the present analysis, F-IBI was introduced to evaluate the health status of the fishery resources in the Beibu Gulf. We accordingly modified the metrics to suit this marine environment. Unlike the more established applications of the IBI to fish communities in rivers and lakes [9,13,15,17,19,25,26], we did not use the metrics of reproductive guild, tolerant species, or native species. Because most marine fish species produce planktonic eggs, we also considered reproductive guild unsuitable for the Beibu Gulf. Furthermore, because the gulf is relatively open water and has a broad connection to the South China Sea, we also considered that the metrics of sensitive species, tolerant species, and native species are likewise unsuitable for biotic integrity assessment here. To better evaluate the status of fishes in this gulf, we added two metrics pertaining to 12 commercial demersal species that are traditionally fished more intensively than other fishes. We believe that paying more attention to their stock status could give us a better understanding of the quality of the fish community and the effectiveness of the fishery management measures at present. Initially, we also considered the density percentage of the zoobenthic-feeding species and the numbers of both the dominant species and the pelagic species, but we did not include these metrics in the final index system because of their subtle interannual variations. Therefore, we selected the indicators that best reflect the current situation of fish in the Beibu Gulf to construct the fish biotic integrity index system.

F-IBI evaluation results showed that the quality of the fishery resources in the Beibu Gulf decreased from 1962 to 2017, which was similar to the results obtained by Chen et al. [27] and Su et al. [28] using other evaluation methods. The biotic integrity of fish in the Beibu Gulf has declined drastically, likely because of overexploitation, climate change, and environmental damage. However, overfishing might have had the greatest impact. At the end of the 1970s, with increases in the numbers and horsepower of fishing boats [29], and continuous improvements in fishing technology (such as application of navigators and fish-finders), marine fishery production in the South China Sea continued to increase. By the 1980s, fish stocks in the Beibu Gulf started to decline [30], while the fishing power in this area was almost still increasing [2]. For the sustainable use of the fishery resources, China's central and local governments adopted a series of fishery management and protection measures, such as a fishing boat licensing system, dual control of the number and horsepower of fishing boats, stock enhancement, and establishment of marine protected areas. A summer moratorium on fishing has been effectively implemented in the northern South China Sea since 1999. Several studies have concluded that the summer moratorium plays an important role in the protection of juvenile fish stocks and effectively alleviates the trend of stock decline in the Beibu Gulf [30–32]. However, some studies found that the effect of the fishing moratorium was limited to that year and could not reverse the trend of resource deterioration [33–36], which was similar to our evaluation results. This also indicated that the evaluation results of the biotic integrity index are credible.

Over the recent years, the total fish catch density in the Beibu Gulf has fallen by 67%, and that of the 12 traditional commercial demersal fishes declined even more sharply, by 87%. This indicates that the fishery stock has been severely overfished, and the high trophic level commercial species traditionally fished are relatively depleted. Assessments found that the species of *P. macracanthus* [37], *E. cardinalis* [38], *D. maruadsi* [39], *N. bathybius* [40], and *P. microcephalus* were overfished in the Beibu Gulf [41]. Zhang et al. [42] found that overfishing in the Beibu Gulf is very serious; 86.7% of the 30 target stocks had suffered from growth overfishing and 83.3% had been overexploited or fully exploited. The fish community in the Beibu Gulf has also changed dramatically, manifesting in altered dominant species, the decrease in nektonic-feeding species biomass, and the increase in pelagic fish proportion. In addition, the number of species recorded by the otter trawl surveys decreased progressively. In 1962, the major dominant species were the large-sized demersal species *L. sanguineus*, *T. myops*, *Pomadasy hasta*, and *N. virgatus*. By 2016–2017, the dominant species were the pelagics *Decapterus maruadsi*, *T. japonicus*, and *P. anomala*, and the small-sized *A. japonicum*, *L. bindus*, and *Apogonichthys ellioti*. The dominant species changed from

species with a larger size, higher value, long life span, high trophic level, and late sexual maturity to species with a smaller size, lower value, short life span, low trophic level, and early maturity [3,4,43,44]. Similar trends have been reported in other regions [45–49]. By using the generalized additive model, Wang et al. [50] found that overfishing was the main driver of sharp declines in demersal fish stocks, with high-value species being replaced by low-value ones, while climate change may be the main driving factor for the increase in pelagic fish in the Beibu Gulf.

All the signs indicate that the biotic integrity of fish in the Beibu Gulf has been steadily deteriorating. However, the F-IBI score for 2006 showed a slight improvement. This may be related to the implementation of a series of fishery management measures since the late 1990s, including a summer moratorium, fishing boat decommissioning, and the "zero growth" policy in fishery production [51]. Nevertheless, fishing intensity from Vietnam has greatly increased in recent decades [52,53], resulting in a further decline in the F-IBI. Therefore, to prevent further deterioration of the biotic integrity, a joint effort from China and Vietnam is needed in implementing fishery management measures, such as reducing fishing capacity, increasing mesh sizes of fishing gears, and protecting fish spawning and nursery grounds. After the implementation of relevant measures, F-IBI can be used to measure the conservation effect, so as to better control the executive strength of management measures.

## 5. Conclusions

The results of the research suggest that the fish community in the Beibu Gulf has been seriously impacted by anthropogenic disturbances, especially overfishing. In this paper, four categories and eight metrics were used to construct the biotic integrity of the fish evaluation index system (F-IBI), and the evaluation results showed that the biotic integrity of fish has significantly deteriorated over recent years. Among eight metrics, the total fish catch density and the catch density of traditional commercial demersal fish species decreased most obviously. This study's use of F-IBI data provides a more explicit description of the status of the gulf's fishery resources, which could better guide the management and utilization of these resources. In order to make the evaluation effect of the index better and more comparable, the survey time, survey area, and survey methods should be as similar as possible.

**Author Contributions:** L.S., Z.C. and Y.Q. conceived and initiated the study and led the writing of the manuscript. Y.X., M.S. and K.Z. participated in the investigation and sample processing. All authors have read and agreed to the published version of the manuscript.

**Funding:** This work was supported by the Key Special Project for Introduced Talents Team of Southern Marine Science and Engineering Guangdong Laboratory (Guangzhou) (GML2019ZD0605), Financial Fund of the Ministry of Agriculture and Rural Affairs (NFZX2021), Key Research and Development Project of Guangdong Province (2020B1111030001), and the Central Public-Interest Scientific Institution Basal Research Fund (2020TD05 and 2021 SD01).

**Institutional Review Board Statement:** The animal study protocol was approved by the South China Sea Fisheries Research Institute Animal welfare committee (protocol code 2019/002 and 1 March 2019).

**Data Availability Statement:** The datasets presented in this study came from the fishery resources surveys over the years. The data are true and reliable. The data used to support the findings of this study are available from the corresponding author upon request.

**Acknowledgments:** We are grateful to all colleagues for their great efforts in the data collection. We also thank Cynthia Kulongowski who edited the language of a draft of this manuscript.

**Conflicts of Interest:** The authors declare no conflict of interest.

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
