# Peer review of "Long-Term Change of a Fish-Based Index of Biotic Integrity for a Semi-Enclosed Bay in the Beibu Gulf"

_fishes, doi:10.3390/fishes7030124_

Round 1

Reviewer 1 Report

This paper showed long-term changes in the demersal fish community of the Beibu Gulf where Chinese and Vietnamese fishing vessels operated. and recognizes the value of the data. In particular, it uses research vessel data rather than fisheries information, which I think is useful as a scientific paper. However, I think there are many improvements, such as the lack of corrections to the data from the 1960s and problems about dietary groups. Please make sufficient corrections.

51-65 IBI proposed by Karr is mainly used in sea areas where there are few major fish species and lakes. I think it is necessary to explain whether it is applicable in the research areas of this study.

81 Catch rate is not suitable. I recommend “abundance index”.

85 This is not a correction for various fishing gears. This calibration is the basic formula of the swept area-density method. Please describe the correction of fishing efficiency. ..

117- (table 3) Fish species composition after 1998 has completely replaced from 1962. This means a considerable change in the fish composition and fish community, rather than increase / decrease of each fish species. The 1962 mesh size is not listed, but this may cause this change. It is so serious problem.

142-147 I have an objection about the dietary group. The main demersal fishes of bottom trawl is benthos eater. Bycatch pelagic fishes are nekton or plankton eater. I think the detritus eater is rare. We can not judge it due to no description of dietary group in Table 3. The dietary group I commented are showed in 185-188, but why didn't you do so?

Figure 1 (left) should be an original map.

Author Response

This paper showed long-term changes in the demersal fish community of the Beibu Gulf where Chinese and Vietnamese fishing vessels operated. and recognizes the value of the data. In particular, it uses research vessel data rather than fisheries information, which I think is useful as a scientific paper. However, I think there are many improvements, such as the lack of corrections to the data from the 1960s and problems about dietary groups. Please make sufficient corrections.

Reply:Thank you for your comments. Through literature review, we found that the net size of fishing gear in the 1960s was 40mm, which is not much different from our current 39mm, and the sampling method is the same, so the data is comparable. For dietary analysis, we used all fish, and classified into three major groups, nektonic, zoobenthos and planktivorous-detritus. There are too many species, so their dietary types are not listed in the table.

51-65 IBI proposed by Karr is mainly used in sea areas where there are few major fish species and lakes. I think it is necessary to explain whether it is applicable in the research areas of this study.

Reply:IBI is widely used in rivers and lakes, and there have been cases at sea area Laizhou Bay (Lin et al, 2020). We have added this literature to the corresponding paragraph in the introduction.

Lin, Q.; Yuan, W.; Shan, X.J.; Li, Z.Y.; Wang J. Evaluation on Biological Integrity of Fish in Laizhou Bay. J. Hydroecology 2020, 42, 101-106.

81 Catch rate is not suitable. I recommend “abundance index”.

Reply:The Catch rate in this paper is finally expressed as Catch density. We use Catch density to represent fish abundance.

85 This is not a correction for various fishing gears. This calibration is the basic formula of the swept area-density method. Please describe the correction of fishing efficiency. ..

 Reply:In different survey periods, our survey types and methods are almost the same, and according to the data checked, the size of fishing gear mesh in 1960s is 40 mm almost the same as ours now 39 mm. Considering the availability of relevant data, it is reasonable to use swept area-density method to calculate catch density and compare resources in different periods.

117- (table 3) Fish species composition after 1998 has completely replaced from 1962. This means a considerable change in the fish composition and fish community, rather than increase / decrease of each fish species. The 1962 mesh size is not listed, but this may cause this change. It is so serious problem.

Reply:Table 3 describes the proportion of dominant species in the total catch density over the years. It is obvious from the table that the species composition in the Beibu Gulf has undergone great changes.  With the advice of another expert, we have added more information of related species in this area.  It is a pity that the original record of the size of the mesh was not found in the 1962 survey.  However, we found some relevant information in a literature. We consulted experts in fishing tackle and fishing law, and checked the Guangdong province Marine fishing gear and fishing law survey report prepared in 1985, and speculated that the size of the fishing gear cod end mesh in this age was 40 mm.  

Wang Y. M. Production reform of bottom trawl fishery of South China Sea after the founding of new China. Journal of Zhanjiang Fisheries Collage, 1995, 15(1):95-97.

142-147 I have an objection about the dietary group. The main demersal fishes of bottom trawl is benthos eater. Bycatch pelagic fishes are nekton or plankton eater. I think the detritus eater is rare. We can not judge it due to no description of dietary group in Table 3. The dietary group I commented are showed in 185-188, but why didn't you do so?

Reply:We take all fish species into consideration in our dietary analysis, while Table 3 showed only dominant species. Therefore, we did not list the diet types of these species in the table 3. When we counted the feeding habits, we also found that detritus feeding species accounted for a small proportion, so we classified them into planktonic feeding species with a relatively low trophic level.  At the same time, we found that benthic feeding species accounted for a high proportion of the total catches, but their inter-annual changes were very small.  The feeding habits of nekton are generally of high trophic level and their annual variation is relatively large. Therefore, we focused on two groups of high and low trophic level, which varied relatively great.  

Figure 1 (left) should be an original map.

Reply:Thanks for your comment. We have removed this map and just used the part we made ourselves. 

Reviewer 2 Report

Please see document attached

Author Response

This paper presents the use of an index of biotic integrity to assess the health of the Beibu Gulf marine ecosystem. Although this work is interesting, I am not sure that as currently written this manuscript properly convey the importance of the results found. While I believe the analysis presented in this paper is sound, the authors need to do a better job at explaining why their analysis is relevant for the management of the gulf fishes and at detailing more clearly how their results can inform resource managers. I also suggest to add some additional statistical analysis to improve the accuracy of the results presented. I provide more specific suggestions below.

Reply:Thank you for your comments. We have added the trend significance analysis and added it to this paper.

Specific comments:

Line 46: Start a new sentence. “In particular, demersal catch composition showed a changing trend from…”

Reply:Thanks for your comment. We have revised this part to use “In particular, demersal catch composition showed a changing trend from…”.

Line 48: Name references need to be removed.

Reply:Name references has been removed.

Line 48-49: reformulate.

Reply:This sentence has been rephrased. “How to better evaluate the decline of the resources is an urgent problem to be solved in order to guide fisheries development and conservation effectively.”

Line 51: Year reference needs to be removed.

Reply:Year reference has been removed.

Lines 53-57: Which organisms were studied? Be more specific.

Reply:The research object of these literatures was fish.

Lines 66: replace by “ an IBI index suitable for assessing the changes in”

Reply:Thanks for your comment. We have replaced it by “ an IBI index suitable for assessing the changes in”.

Line 68-70: Replace by “ The index scores were developed based on trawl surveys conducted for 5 years over the period 1962 to 2017. Temporal trends in the IBI scores were determined … This study provides….”

Reply:Thanks for your comment. We have replaced the relevant statement with “ The index scores were developed based on trawl surveys conducted for 5 years over the period 1962 to 2017. Temporal trends in the IBI scores were determined … This study provides…”.

Line 79: Start with “ To use the catch data from the different sampling periods, ….”

Reply:Thanks for your comment. We have replaced this part of the statement with “ To use the catch data from the different sampling periods”.

Line 80: How many sites did you use?

Reply:The detailed situation of survey stations is described in Table 1. There were 25 stations in 1962, 43 stations in 1998, and 38 stations from 2006 to 2017, and we have added the station information here.

Lines 98-100: How did you choose those metrics/attributes? Please detail more. Also did you estimate the correlation between metrics to eliminate redundancy?

Reply:Thanks for your comment. How we choose these parameters is described in detail in the first paragraph of the discussion section. In the selection process, we combine literature and research practice to leave the indicators that can best reflect resource changes, so we do not give much consideration to using statistical methods to select indicators here.

Line 100: end of the sentence is not clear. Reformulate

Reply:We have reformulated,and marked it in red.

Line 105: Do you have literature reference on demersal fish values?

Reply:Relevant references are available, such as Chen, Z.C., Liu, J.X., 1982. Commercial fishes of the South China Sea. Guangdong Scientific and Technological Press, Guangzhou, 266pp. (in Chinese).

Line 106-110: I would write:” 1962 conditions were considered the reference point and each metric was scored again 1962 data. For metrics that were expected to decrease in response to interference (i.e.,fishing pressure), S = …, and for metrics expected to increase S=…”

Reply:Thanks for your comment. We have made the modification according to your opinion.

Table 2: I would name each metric such as M1, M2, M3….

Reply:We have made the modification according to your opinion.

Section 3.1.: Could you give more information on the dominant species, such as demersal vs pelagic?Importance for a given fishery? Ecological role?

Reply:Thanks for your comment. We have added some descriptions of the dominant species to better show the transformation of the dominant species. Because species are important to fisheries, they can vary from time to time. In the 1960s, large fish like red snapper Lutjanus sanguineus might have been important, but as resources decline, now they, small pelagic fish like Decapterus maruadsi and Trachurus japonicus, become more important, so we do not emphasize it. 

Line 131-133: Should be included in the methods section when explaining that 1962 was used as the reference point.

Reply:Thanks for your comment. This section has been moved to the methods section.

Line 133-139: Please support the described changes in IBI score with a statistical analysis testing the significance of increasing and decreasing trends. First the data need to be tested for Normality and Variance homogeneity and based on the results either use some ANOVA or Kruskal-Wallis test.

Reply:Thanks for your comment. Since there are only five periods of data and the amount of data is small, it is not suitable for normal analysis. Therefore, we conducted a significance analysis on its changing trend, and the analysis showed that its changing trend showed a significant decline.

Figure 2: I would color the histogram by IBI score category: excellent vs fair.

Reply:Thanks for your comment. We used dark and light colors to distinguish excellent from fair in the Figure 2.

Section 3.3: this section should be moved before section 3.2 to present the changes in each metrics before final changes in IBI. Also the paper would benefit from having a table with each metrics values for each time period and a statistical test performed to assess whether changes in metrics were significant over time. See comment above for details on stat methods.

Reply:Thanks for your comment. We deliberated for a long time on the order of placing these two paragraphs, and finally arranged the order of F-IBI index in the first place, and the change trend of metrics in the second place. The reason was to consider the whole first and then the part. We first saw the change trend of the whole, and then described in detail which metrics caused the change trend. Each metrics values for each period shown in Figure 3, and we have added an analysis of their trend significance. 

Figures 3, 4, 6 and 6: Combine them in a multifaceted figure.

Reply:Thanks for your comment. Figures 3, 4, 5 and 6 have been combined in the same figure 3.

Line 172-182: This needs to be introduced in the methods section.

Reply:Thanks for your comment. This section is a discussion of the method, why we do it, has a discussion nature, so we put this section at the beginning of the discussion section. 

Line 185-189: This also should be moved to the methods.

Reply:Thanks for your comment. This section is a discussion of the method, why we do it, has a discussion nature, so we put this section at the beginning of the discussion section. 

Line 192: What do you base this statement on? Has there been a study looking at the impact of climate change and or environmental damage vs fishing pressure? If not I would reformulate and describe how based on previous study it is known that overfishing has a strong impact of fish communities.

Reply:Thanks for your comment. This statement is based on an analysis of the literature, and then it was explained. The increasing number and power of fishing boats, the improvement of fishing technology, and the increasing production of Marine catch etc. indicated that overfishing is the main reason for the decline of fishery stocks and the changes of community. 

Line 206-207: Re-emphasize the findings of this study and explain what new information does it bring compared to previous cited work.

Reply:Thanks for your comment. The research results cited in the literature are mainly descriptive, a description of the phenomenon, but there is no good measurement of the decline of fishery resources to what extent and how much, and these are the problems we want to solve in this study.  

Lines 216-224: Instead of repeating the results found this section would benefit from more speculation on how potential mechanisms that led to those species changes and ultimately to a decrease in IBI.

Reply: The description of dominant species in this paragraph also reflects the reasons for the decline of IBI index. The change of large species to small species means that the number of fish with nekton diet will decrease and the IBI index will decrease. The increase of pelagic fish also reflects the increase of planktonic diet fish, which will also lead to the decline of IBI index. Therefore, we will write later that all these signs indicate that biotic integrity of fish in the Beibu Gulf has been steadily deteriorating. 

Line 226: Regions of China or in the world? Be more specific.

Reply:This paragraph describes only the Beibu Gulf of China, and it is marked at the end of the sentence.

Lines 236-239: This section needs to be developed and explain how the IBI work done here could help develop management or conservation measures.

Reply:Thanks for your comment. We have added relevant content “After the implementation of relevant measures, F-IBI can be used to measure the implementation effect, so as to better control the implementation of relevant measures. ”.

Reviewer 3 Report

The authors contribute a solid data volume, which can be consolidated to extract serious results. The manuscript has good citation potential and should be published. Nevertheless, it has not obtained a final view.

Sampling methodology and adaptations are well described and followed, but some small omissions have been noticed considering this part: it is not clear how species have been determined (atlas, book, identification key, etc.) What about eventual older names from previous samplings, are they updated?

The description of the calculations leading to certain index values accordingly could be better. Should samplings from the 60s be considered as referent or also show degradation? (historical + current data, but expert judgment is lacking). The ecological Quality Ratio (EQR) is calculatable. A decrease in commercial fisheries has been noticed in the area, which is also reflected in the EQRs. But there is no clear (at least statistical) connection between pressure and response (fishing effort/fish community deterioration). Other pressures, e.g. pollution, contribute to the cumulative effect.

Why borders between status classes have been determined as proposed? The border between good and moderate status is crucial and should be carefully set.

The temporal dimension of the proposed assessment system is visible, but what about the spatial? Is the index applicable to broader marine territories e.g. around Hong Kong?

There is no significant information, concerning similar cases in discussion (assessment systems in marine environments).

In conclusion, a valuable contribution is hidden behind general statements. The authors use a more empirical pathway – which is right! – but could be supported by further statistical analyses. The manuscript could be split into two different ones: 1. Fish community temporal alterations and 2. Construction of a fish-based ecological assessment system. They are hereby mixed and focus is rather given on the alterations, than on the assessment. If authors intend to publish it joined as it is, much more effort and volume will be necessary, so as to cover every issue.

The reviewer kindly proposes to the authors for becoming acquainted with European good practices concerning fish fauna assessment and especially the Intercalibration process. Manuals could be found online as well, and many ideas could be borrowed, so as to increase the strength of the derived results. Nevertheless, the manuscript should pass under certain revision.

Author Response

The authors contribute a solid data volume, which can be consolidated to extract serious results. The manuscript has good citation potential and should be published. Nevertheless, it has not obtained a final view.

Sampling methodology and adaptations are well described and followed, but some small omissions have been noticed considering this part: it is not clear how species have been determined (atlas, book, identification key, etc.) What about eventual older names from previous samplings, are they updated?

Reply:Thanks for your comment. Species identification is done by experts from the same institution, so there is some continuity. Some species do have different Chinese names, but we use their Latin names.  

The description of the calculations leading to certain index values accordingly could be better. Should samplings from the 60s be considered as referent or also show degradation? (historical + current data, but expert judgment is lacking). The ecological Quality Ratio (EQR) is calculatable. A decrease in commercial fisheries has been noticed in the area, which is also reflected in the EQRs. But there is no clear (at least statistical) connection between pressure and response (fishing effort/fish community deterioration). Other pressures, e.g. pollution, contribute to the cumulative effect.

Reply:Thanks for your comment. Fishery resources were in good condition at the 1960s, and relevant studies have confirmed this.  Therefore, it is feasible to use this period as a reference point. This paper mainly intends to discuss the resource status from the aspect of fish biological integrity. Next, we will also consider the factors that lead to the decline of fishery resources, such as fishing, pollution and other pressures on fishery resources. We are looking for ways to better measure these pressures. 

Chen, Z.Z.; Qiu, Y.S.; Jia, X.P.; Huang, Z.R.; Wang, Y.Z. Structure and function of Beibu Gulf ecosystem based on Ecopath model. J. Fish. Sci. Chin. 2008, 15, 460–468.

Chen, Z.Z.; Qiu, Y.S.; Xu, S.N. Changes in trophic flows and ecosystem properties of the Beibu Gulf ecosystem before and after the collapse of fish stocks. Ocean Coast. Manage. 2011, 54, 601–611.

Why borders between status classes have been determined as proposed? The border between good and moderate status is crucial and should be carefully set.

Reply:Thanks for your comment. The boundary value is a key value, and we have considered it carefully during the determination. We determined the evaluation grade and classification standard by referring to relevant literature, and combined with our evaluation results, it is reasonable to divide this grade into five grades.  

Lin, Q.; Yuan, W.; Shan, X.J.; Li, Z.Y.; Wang J. Evaluation on Biological Integrity of Fish in Laizhou Bay. J. Hydroecology 2020, 42, 101-106.

Moncayo-Estrada, R.; Lyons, J.; Escalera-Gallardo, C.; Lind, O.T. Long-term change in the biotic integrity of a shallow tropical lake: A decadal analysis of the Lake Chapala fish community. Lake Reserv. Manage. 2012, 28, 92–104.

The temporal dimension of the proposed assessment system is visible, but what about the spatial? Is the index applicable to broader marine territories e.g. around Hong Kong?

Reply: Thanks for your comment. This is a good idea, if the index is used in different Spaces, so that the status of fishery resources in different regions can be compared and analyzed.  If historical survey data are available in Hong Kong waters, this method can also be used for evaluation. We also plan to use this index in other seas.

There is no significant information, concerning similar cases in discussion (assessment systems in marine environments).

Reply: Since the Marine Environmental Assessment system and the fish biotic integrity assessment index are very different, so we compared the results with similar studies, and their evaluation results were similar to ours.

Chen, Z.Z.; Lin, Z.J.; Qiu, Y.S. Evaluation of sustainability of fisheries resources for South China Sea based on the AHP. J. Natural Resources 2010, 25, 249–257.

Su, L.; Chen, Z.Z.; Zhang K.; Xu Y.W.; Qiu Y.S. Establishment of quality status evaluation system of fisheryresources in Beibu Gulf based on bottom trawl survey data. J. Guangdong Ocean U. 2021, 41, 10–16.

In conclusion, a valuable contribution is hidden behind general statements. The authors use a more empirical pathway – which is right! – but could be supported by further statistical analyses. The manuscript could be split into two different ones: 1. Fish community temporal alterations and 2. Construction of a fish-based ecological assessment system. They are hereby mixed and focus is rather given on the alterations, than on the assessment. If authors intend to publish it joined as it is, much more effort and volume will be necessary, so as to cover every issue.

Reply:Thanks for your comment. We have appropriately added F-IBI constructs at the conclusion. “In this paper, four categories and eight metrics were used to construct the biotic integrity of fish evaluation index system (F-IBI), and the evaluation results showed that the biotic integrity of fish significantly deteriorated for over recent years.”

The reviewer kindly proposes to the authors for becoming acquainted with European good practices concerning fish fauna assessment and especially the Intercalibration process. Manuals could be found online as well, and many ideas could be borrowed, so as to increase the strength of the derived results. Nevertheless, the manuscript should pass under certain revision.

Reply:Thanks you for your advice. We have also seen papers from researchers in European countries like Norway, and we have considered the parameters they choose. Maybe some details are not so comprehensive, we will continue to strive to learn more about the relevant knowledge to improve our work, so that we can do better in future fishery resource evaluation. 

Round 2

Author Response

Line 56-62: Too many examples to get to the same conclusion that IBI is a useful biomonitoring tool. One would suffice.

Reply: Thank you for your comment. We have deleted lines 56-60, leaving only one example.

Line 67-69: I am not sure I understand what is the difference between the first (first part of the sentence) and second (second part) objectives. It needs to be explained better.

Reply: Thank you for your comment. The expert revision opinion already contains the two meanings that we want to express, the second half sentence appears redundant, so we have deleted it. And modified to “The purpose of this study was to develop an IBI index suitable for assessing the changes in the integrity of the fishery resources in Beibu Gulf in different years.”.

Line 107: beibu should be written with capital B: Beibu. Also please add a reference to this statement.

Reply: Thank you for your comment. We have modified to “Beibu”. And marked in light blue.

Line 138: Replace “The study” by “We”. The authors make the assumptions.

Reply: Thank you for your comment. We have replaced “The study” by “We”.

Line 139: add “then” after the comma and before “slightly increased”.

Reply: We have added “then” after the comma and before “slightly increased”.

Line 143-144: More details on the trend analysis performed are needed. A statistical test section should be added in the methods to provide details on the tests used before the result are presented.

Reply: Thank you for your comment. We have added “The values of F-IBI and metrics were plotted against sequence of years, and the statistical significance of their trend over year was tested using regression analysis in Excel 2016.” in the methods.

Section 3.3: Same comment than above the statistical tests used to study the trend patterns need to be detailed in the methods.

Reply: We have added “The values of F-IBI and metrics were plotted against sequence of years, and the statistical significance of their trend over year was tested using regression analysis in Excel 2016.” in the methods.

Figure 3 legend: Figure 3c caption is the same as 3b. Likely a mistake.

Reply: Thank you for your comment. We have modified to “the percentages of density of pelagic species and dominant species”.

Line 243-245: This sentence needs to be reformulated, too many word repetitions.

Reply: Thank you for your comment. This sentence has been amended to “After the implementation of relevant measures, F-IBI can be used to measure the conservation effect, so as to better control the executive strength of management measures. ”
